

# Learning lattice quantum field theories
# with equivariant continuous flows

**Mathis Gerdes[1⋆°], Pim de Haan[2,3†°], Corrado Rainone[3],**
**Roberto Bondesan[3] and Miranda C. N. Cheng[1,4,5]**

**1** Institute of Physics, University of Amsterdam, the Netherlands
**2** QUVA Lab, University of Amsterdam
**3** Qualcomm AI Research,[1] Qualcomm Technologies Netherlands B.V.
**4** Korteweg-de Vries Institute for Mathematics, University of Amsterdam, the Netherlands
**5** Institute for Mathematics, Academia Sinica, Taipei, Taiwan

⋆ m.gerdes@uva.nl , † pim.de.haan@uva.nl

## Abstract

We propose a novel machine learning method for sampling from the high-dimensional probability distributions of Lattice Field Theories, which is based on a single neural ODE layer and incorporates the full symmetries of the problem. We test our model on the $\phi^4$ theory, showing that it systematically outperforms previously proposed flow-based methods in sampling efficiency, and the improvement is especially pronounced for larger lattices. Furthermore, we demonstrate that our model can learn a continuous family of theories at once, and the results of learning can be transferred to larger lattices. Such generalizations further accentuate the advantages of machine learning methods.



## Contents

---

°These authors contributed equally to this work.
[1]Qualcomm AI Research is an initiative of Qualcomm Technologies, Inc.

# 1 Introduction

Lattice field theory (LFT) remains the only robust and universal method to extract physical observables from a non-perturbative quantum field theory (QFT). As a result, the applications of such lattice computations are omnipresent in physics. Calculations for LFTs have traditionally relied on Markov Chain Monte Carlo (MCMC) methods. A well-known challenge these sampling methods face is the phenomenon of critical slowing down [1]. When moving towards the critical point or the continuum limit, consecutively generated samples become increasingly correlated as measured by a rapidly increasing *autocorrelation time* (time between statistically independent samples). In practice, this computational inefficiency constitutes a bottleneck for achieving high-accuracy results. While efficient algorithms exist for certain statistical mechanical models — e.g. the Swendsen-Wang algorithm for the Ising and Potts models [2] — a general approach is still lacking. An additional challenge arises from the fact that the entire sampling procedure typically needs to be initiated anew each time the theory is deformed, for instance by changing the UV cutoff (lattice spacing) or the coupling constants.

Rapid progress in machine learning provides new tools to tackle this complex sampling problem, see [3–20] for earlier related works. Flow-based models, in particular, offer a promising new framework for sampling which can mitigate the problem of critical slowing down and amortizes the costs of sampling by learning to approximate the target distribution. Moreover, since they estimate the likelihood of configurations they can be used to estimate some observables which are challenging to estimate with standard MCMC methods [21, 22]. In the flow framework, a parametrized function $f_\theta$ is learned, which maps from an auxiliary "latent" space to the space of field configurations. It is optimized such that an easy-to-sample distribution $\rho$ (e.g. an independent Gaussian) gets transformed, or pushed forward, to approximate the complex distribution given by the action of the theory [23]. This is similar to the idea of trivializing maps [24], except that the map itself is approximated using machine learning methods. Moreover, the capability to incorporate physical symmetries of either the local or global type into the flow can circumvent difficulties of sampling caused by the degeneracies of physical configurations which are related by symmetry actions [25].

The idea of applying a flow model to lattice field theory has been tested in [14] using the so-called real NVP model, and proof-of-concept experiments have been successfully carried out for small lattices. However, significant hurdles remain before a flow-based model can be applied in a physically interesting setup. For instance, it has been argued that these models require a training time that rises exponentially with the number of lattice sites [26]. In practice, it becomes infeasible to attain high acceptance rates on large lattices, while these are precisely the physically interesting regime.

To go beyond proof-of-concept studies, we devise a novel flow model[2] which improves the performance in the following three aspects, when compared to the state-of-the-art baseline model [14, 18, 26]: 1) scalability, 2) efficiency, and 3) symmetries. In more details, we contribute the following. 1) We propose a *continuous flow* model, where the inverse trivializing map $f_\theta$ is given as the solution to an ordinary differential equation (ODE). When applied to the two-dimensional $\phi^4$ theory, our model dramatically improves a key metric, the effective sample size (ESS), to 91% at $L^2 = 32^2$ after about 10 hours of training as shown in Figure 1, in contrast to the 1.4% achieved by the real NVP model after training saturates. Going further beyond what has been reported so far [26], we also obtain good results on lattices with $L^2 = 64^2$ vertices. Moreover, compared to the real NVP flow tested in [26], Figure 2 demonstrates that our continuous flow model is significantly more sample efficient in training and, in particular, displays a significantly less steeply growing training cost as lattice size increases. 2) We demonstrate the ability of our model to learn many theories at once in the following sense. First, we show that training efficiency can be gained by upscaling the network learned for lattice size $L$ to a larger lattice of size $L'$. See Figure 1. Second, we train a family of flows $f_\theta^{(\psi)}$ parametrized by theory parameters $\psi$ within a continuous domain, as used for Figure 4. Once the training is done, sampling is extremely cheap and no particular slowing down is encountered near the critical point. 3) In contrast to the real NVP architecture (cf. [27]), our model is equivariant to all symmetries of the LFT, including the full geometric symmetries of the lattice and the global symmetry of the scalar theory. If not built into the architecture, these symmetries are only approximately learned by the model, as exemplified in Figure 6.

## 2 Lattice quantum field theory

Our goal is to improve the MCMC sampling performance for quantum field theories with scalar fields which possess non-trivial symmetry properties. We will now consider a scalar field theory on a two-dimensional periodic square lattice $V_L \cong (\mathbb{Z}/L\mathbb{Z})^2$. In particular, the *field configuration* is a real function $\phi : V_L \to \mathbb{R}$ on the vertex set $V_L$, and we denote by $\phi_x$ the value at the vertex $x \in V_L$. The theory is described by a probability density

$$p(\phi) = \frac{1}{Z} e^{-S(\phi)}, \quad \text{with action}$$

$$S(\phi) = \sum_{x,y \in V_L} \phi_x \Delta_{xy} \phi_y + \sum_{x \in V_L} m^2 \phi_x^2 + \sum_{x \in V_L} V(\phi_x), \tag{1}$$

where $V : \mathbb{R} \to \mathbb{R}$ is a potential function, $\Delta$ is the Laplacian matrix of the periodic square lattice, and $m^2$ is a numerical parameter. The partition function is defined as

$$Z = \int \prod_{x \in V_L} d\phi_x \, e^{-S(\phi)}. \tag{2}$$

For polynomials $V(\phi)$ of degree greater than two, the normalization factor $Z$ and the moments of $p(\phi)$ are not known analytically, and are typically estimated numerically. The statistical correlations between spatially-separated degrees of freedom of the theory are measured by the *correlation length* $\xi$ (see appendix A for numerical details). As a result, it also controls the difficulty of directly sampling from $p(\phi)$.

We focus here on the case where the potential is invariant to all spatial symmetries of the lattice and has the $\mathbb{Z}_2$ symmetry $V(\phi) = V(-\phi)$. Technically, this family of theories is also

---

[2]An implementation can be found at https://github.com/mathisgerdes/continuous-flow-lft.

known as the Landau-Ginzburg model, and possesses a rich phase diagram with multi-critical points famously described by the unitary minimal model conformal field theories. See [28] for details.

## 2.1 Samplers based on normalizing flows

To correct for the approximate nature of any learned distribution, a Metropolis-Hastings step is applied to reject or accept the generated samples. This guarantees asymptotic exactness without significantly increasing the computational cost, assuming the *acceptance rate* is sufficiently high. Recall that the Independent Metropolis-Hastings algorithm is a flexible method to sample from a target density $p$ given a proposal distribution $q$ [29]. At each time $i$, a proposed sample $\phi'$ drawn from the proposed distribution $q$ is accepted with probability

$$\min\left(1, \frac{q(\phi^{(i-1)})p(\phi')}{q(\phi')p(\phi^{(i-1)})}\right),\tag{3}$$

which depends on the previous sample $\phi^{(i-1)}$. Accepting the proposal means setting $\phi^{(i)} = \phi'$, else we repeat the previous value $\phi^{(i)} = \phi^{(i-1)}$.

Traditionally, the distribution $q$ is painstakingly handcrafted to maximize the acceptance rate. In contrast, the ML approach [30] to this problem is to learn a proposal distribution $q(\phi)$, for example by using a normalizing flow [14]. As mentioned before, in our framework $q(\phi)$ is given by the push-forward under a learned map $f_\theta$ of some simple distribution $\rho$ such as a normal distribution. Since the proposals are generated independently, rejections in the ensuing Metropolis-Hastings step are the only source of autocorrelation.

When samples from $p$ are available, $q$ can be matched to $p$ by optimizing the parameters $\theta$ to maximize the log-likelihood of those samples: $\mathbb{E}_{\phi\sim p}[\log q_\theta(\phi)]$. However, obtaining such samples requires expensive MCMC sampling which is precisely what we would like to avoid. Instead, we aim to minimize the reverse Kullback-Leibler (KL) divergence $\mathrm{KL}(q|p)$, also known as the reverse relative entropy, which uses samples from the model distribution $q$:

$$\begin{aligned}
\mathrm{KL}(q|p) &= \mathbb{E}_{\phi\sim q}\left[\log\frac{q(\phi)}{p(\phi)}\right] \\
&= \mathbb{E}_{z\sim\rho}[\log q(f_\theta(z)) + S(f_\theta(z))] + \log Z\,.
\end{aligned}\tag{4}$$

In the second equality, we have used the flow $f_\theta$ to replace samples from $q$ by $f_\theta(z)$, where $z$ denotes samples of $\rho$. Note that the last term does not contribute to the gradient and we thus do not need to compute the partition function.

Besides the acceptance rate of the above Metropolis-Hastings scheme, the quality of the optimized map $f_\theta$ can be evaluated using the (relative) effective sample size, which indicates the proportion of number of effective samples from the true distribution to the number of proposed samples. Given a set of $N$ proposed samples $\{\phi^{(i)}\}_{i=1}^N$ from $q$, generated using $f_\theta$, the ESS is computed as [18]:

$$\mathrm{ESS} = \frac{\left[\frac{1}{N}\sum_{i=1}^N p(\phi^{(i)})/q(\phi^{(i)})\right]^2}{\frac{1}{N}\sum_{i=1}^N \left[p(\phi^{(i)})/q(\phi^{(i)})\right]^2}\,.\tag{5}$$

## 3 Continuous normalizing flow

In our continuous normalizing flow model, we transform a sample $z \sim \rho$ with an invertible map

$$f_\theta : \mathbb{R}^{L^2} \to \mathbb{R}^{L^2}, \quad z \mapsto \phi\,,\tag{6}$$

defined as the solution to a neural ODE [31] for time $t \in [0, T]$ of the form

$$\frac{\mathrm{d}\phi(t)_x}{\mathrm{d}t} = g_\theta(\phi(t), t)_x, \quad \text{with} \quad z \equiv \phi(0), \ \phi \equiv \phi(T). \tag{7}$$

Here, the vector field $g_\theta(\phi(t), t)_x$ is a neural network with weights $\theta$. The probability of samples transformed by the flow $q(\phi) \equiv p(\phi(T))$, necessary for computing the KL divergence, can be obtained by solving another ODE [31] which contains the divergence of $g_\theta$:

$$\frac{\mathrm{d}\log p(\phi(t))}{\mathrm{d}t} = -(\nabla_\phi \cdot g_\theta)(\phi(t), t), \tag{8}$$

with the boundary condition $p(\phi(0)) = \rho(z)$. For the network architectures of the vector field $g_\theta$ described below, the divergence can be computed analytically. The gradients of the KL divergence are computed with another ODE derived using the adjoint sensitivity method [31].

As mentioned in [32], if the vector field $g_\theta$ is equivariant to the symmetries of the theory and the chosen latent space distribution is invariant under the symmetry action, then the resulting distribution on $\phi$ is automatically invariant. In other words, for a group element $h \in G$ acting on the space of field configurations $\mathbb{R}^{L^2}$, if $\rho(hz) = \rho(z)$ and $h(g_\theta(\phi(t), t)) = g_\theta(h\phi(t), t)$ it follows that $p(h\phi(t)) = p(\phi(t))$. In the next section we show how to construct a fully equivariant vector field $g_\theta$.

## 3.1 Neural ODE architecture

Inspired by equivariant flows used for molecular modelling [32], we propose to construct a time-dependent vector field for the neural ODE featuring pair-wise interactions between the lattice sites and a tensor contraction with a chosen set of basis functions:

$$\frac{\mathrm{d}\phi(t)_x}{\mathrm{d}t} = \sum_{y,d,f} W_{xydf} K(t)_d H(\phi(t)_y)_f. \tag{9}$$

With chosen dimensions $D$ and $F$, the sums of $d$ and $f$ are taken over $1, \ldots, D$ and $1, \ldots, F$, respectively. Here, $H : \mathbb{R} \to \mathbb{R}^F$ is a basis expansion function for the local field values $\phi_y$, $K : [0, T] \to \mathbb{R}^D$ is a chosen time kernel, and $W \in \mathbb{R}^{L^2 \times L^2 \times D \times F}$ is a learnable weight tensor.

Our choice for the basis function is $H(\phi)_1 = \phi$ and $H(\phi)_f = \sin(\omega_f \phi)$ for $f \in \{2, ..., F\}$, namely a linear term combined with a sine expansion where $\omega \in \mathbb{R}^{F-1}$ are learnable frequencies, inspired by Fourier Features [33]. This choice of basis functions is made in order to respect the global symmetry $\phi \mapsto -\phi$, as is done in [21]. In particular, the cosine function is prohibited by the symmetry. The functional dependence of (9) on $\phi$ is also chosen so that the divergence of the vector field, necessary for computing the density of the resulting distribution $q$, can be easily computed analytically. Other choices of basis functions which respect the global symmetry are conceivable. Odd polynomials, in particular, may appear to be a natural choice; however, for these, we have found the training dynamics to be unstable. While our preliminary experiments have shown the trigonometric basis above to perform well for the $\phi^4$-theory in our range of coupling constants, the optimal choice will likely depend on the particular application.

The time basis functions $K : [0, T] \to \mathbb{R}^D$ are chosen to be the first $D$ terms of a Fourier expansion on the interval $[0, T]$. This is chosen to allow for simple time-dependent dynamics.

The learnable weight tensor $W$ is initialized to 0, so that the initial flow is the identity map. Linear constraints on the tensor imposed by equivariance with respect to spatial symmetries reduces the number of independent entries of the weight tensor. The spatial symmetry group of the periodic lattice $(\mathbb{Z}/L\mathbb{Z})^{\times 2}$ is

$$G = C_L^2 \rtimes D_4, \tag{10}$$

Table 1: ESS, MCMC acceptance rate and observables $\xi$ and $\chi_2^{(1)}$ for the flow models trained with different lattice sizes $L$ and the corresponding $\lambda$ tuned to give $L/\xi = 4$ [34].

| $L$ | $\lambda$ | ESS | MCMC | $L/\xi$ | $\chi_2^{(1)}$ |
|---|---|---|---|---|---|
| 6 | 6.975 | 99.759(7) | 97.503(4) | 3.968(5) | 1.064(2) |
| 12 | 5.276 | 98.54(2) | 93.367(7) | 3.981(5) | 4.129(6) |
| 20 | 4.807 | 96.63(4) | 89.77(1) | 4.039(5) | 10.59(2) |
| 32 | 4.572 | 91.07(8) | 83.15(2) | 4.031(5) | 25.53(4) |
| 64 | 4.398 | 66(5) | 64.96(2) | 4.012(4) | 91.6(2) |

the semi-direct product of two cyclic groups $C_L$ acting as translations, and the Dihedral group $D_4$ generated by 90-degree planar rotations and the mirror reflection. $G$-equivariance dictates that $W_{xydf}$ for a given $d$ and $f$ only depends on the orbit of the pair $(x, y)$. Specifically, for any element $g$ of the symmetry group $G$ acting on a pair of lattice points as $(x, y) \mapsto (gx, gy)$, we have $W_{gx,gy,df} = W_{xydf}$ for all values of $x$, $y$, $d$, and $f$. Explicitly, since we can use $C_L^2$-equivalences to move $x$ to the (arbitrarily chosen) origin given any pair $(x, y)$, and for most of these pairs there are eight $y'$ satisfying $(x, y) \sim (x, y')$ under $D_4$ equivalance, the number of free parameters per $d$ and $f$ grows like $L^2/8 + O(L)$ as opposed to $L^4$ without symmetry constraints.

Finally, we observe that the following factorization of the matrix leads to better training results:

$$W_{xydf} = \sum_{d'f'} \tilde{W}_{xyd'f'} W_{d'd}^K W_{f'f}^H \,, \tag{11}$$

with matrices $W^K \in \mathbb{R}^{D' \times D}$ and $W^H \in \mathbb{R}^{F' \times F}$. The bond dimensions $D'$ and $F'$ are hyperparameters of the model.

## 3.2 A theory-conditional model

A small change in the parameters of the theory typically leads to a small deformation of its associated distribution, which is reflected in our observation that the network trained for a specific theory still results in a relatively high value of ESS even when applied to a nearby theory with slightly altered parameters. This motivates the theory-conditional model and the transfer learning between lattice sizes, as discussed below and in the following section, respectively.

Instead of a single action $S$, here we consider a set of theories $\mathcal{M}$ on the lattice. Each theory $\psi \in \mathcal{M}$ is specified by the action $S_\psi : \mathbb{R}^{L^2} \to \mathbb{R}$, and we can then learn a single neural network to generate samples for all these theories. We do this via an additional basis expansion function $J : \mathcal{M} \to \mathbb{R}^A$ and a new index to each of the terms in the factorized $W$ tensor: instead of (11), we let

$$W_{xydf}(\psi) = \sum_{d'f'abc} \tilde{W}_{xyd'f'a} J(\psi)_a W_{d'db}^K J(\psi)_b W_{f'fc}^H J(\psi)_c \,. \tag{12}$$

By choosing a distribution $r(\psi)$ over the theories, the parameters of the model are optimized by maximising the reverse KL divergence averaged over the theories: $\mathbb{E}_{\psi \sim r} \mathrm{KL}(q_\psi | p_\psi)$, where $q_\psi$ is the push-forward distribution using to the map solving the ODE (9) with the conditional weight matrix (12) and $p_\psi$ is the Boltzmann distribution of action $S_\psi$.

## 3.3 Transfer-learning between lattice sizes

The parameters $W$ of a model that is trained on a square lattice with side length $L$ may be used as the initial parameters $W'$ for training a model on a larger lattice with length $L'$. This is an

example of transfer learning or curriculum learning [35] and may lead to a speedup compared to training directly on the lattice of length $L' > L$. Roughly speaking, to transfer we embed the kernel $W$ into the kernel $W'$ for the larger lattice. The frequencies $\omega_f$ are kept fixed during the transfer. After initializing the parameters this way, the model can be further trained on the lattice of length $L'$.

More concretely, to embed the kernel $W$ into the kernel $W'$ for the larger lattice, we consider a pair of points $(x, y)$ in the group of two-dimensional cyclic translations with period $L$, $C_L^2$, and a pair of points $(x', y')$ in the group $C_{L'}^2$. We define identify $\tau = y - x \in C_L^2$ with $\tau = [\tau_1, \tau_2]$ where $\tau_i \in \{-\lfloor\frac{L}{2}\rfloor, \ldots, \lfloor\frac{L-1}{2}\rfloor\}$, and similarly $\tau' = y' - x' = [\tau_1', \tau_2']$ with $\tau_i \in \{-\lfloor\frac{L'}{2}\rfloor, \ldots, \lfloor\frac{L'-1}{2}\rfloor\}$. To transfer, we set as initial values

$$W'_{x'y'af} = W_{xyaf} N_L(\tau)/N_{L'}(\tau'), \tag{13}$$

if $\tau = \tau'$, and zero otherwise. The scaling factor

$$N_L(\tau) = |\{R\tau R^{-1} \in C_L^2 \mid R \in D_4\}|, \tag{14}$$

is the number of sites reached by rotating/mirroring $y$ around $x$ in a periodic lattice of length $L$, where $RtR^{-1}$ denotes the action of $R$ on $t$. It accounts for the fact that weights may be used for more pairs in the larger lattice, as illustrated in Figure 7 in the appendix.

# 4 Numerical tests

For our experiments, we consider the $\phi^4$ theory, which has an action as in equation (1) with potential

$$V(\phi) = \lambda \sum_x \phi_x^4, \tag{15}$$

specified by the coupling constant $\lambda$. This can either be fixed, giving a single theory target, or we can aim to learn for a range of values, while adding $\lambda$ as an input to the network. If not specified otherwise, the ODE's defined by the vector field models are solved using a fourth-order Runge-Kutta method (RK4) with a fixed number of 50 steps. See section C for a discussion of the discretization error.

## 4.1 Single theory training

In the first experiment, we evaluate the ability of our model as defined in equation (9) to learn to generate samples from a single theory, i.e. for a fixed value of $\lambda$. We test this on lattices with length $L$ varying between 6 and 64, where the theory parameter $m^2$ is held fixed to be $m^2 = -4$, and $\lambda$ is tuned so that the correlation length $\xi$ approximately equals to 1/4 of the lattice side length. We choose $F = 50$ for the number of field basis functions, $D = 21$ for the number of time kernels, and $F' = 20$, $D' = 20$ for the bond dimensions of the respective factorization matrices. The parameters are optimized by gradient descent, where the expectation in the loss of equation (4) is approximated by averaging over 256 generated samples. Further implementation details can be found in the appendix. To assess model quality, we use two different but related metrics, the ESS and the MCMC acceptance rate. We remind the reader that the ESS is a quantity closely related to the acceptance rate. It controls the variance of the mean of $N$ correlated random variables, which scales as $(\text{ESS} \times N)^{-1/2}$. In particular, an ESS of 100% indicates perfectly independent samples and is attained when the target and the proposal distribution coincide. In Figure 1 we report the ESS values against the training time, computed as a moving average over 100 training steps, with the real NVP baseline for

comparison, which is based on [18] and was trained until the ESS saturates (see appendix for details). For each solid line, the corresponding coupling $\lambda$ is chosen such that the correlation length is approximately $L/\xi = 4$, as shown in Table 1. Our model attains much larger values of the ESS and acceptance rates in much shorter times, for all sizes $L$, despite the fact that each of its training steps involves integration of an ODE. Table 1 shows the final ESS and MCMC acceptance rates as well as $L/\xi$ and the two-point susceptibility $\chi_2^{(1)}$, computed with MCMC using the trained flows. Note that the correlation length computed with our flow-based MCMC is indeed comparable with that computed using the traditional MCMC method. Details on how $\xi$ and the two-point susceptibility are estimated given MCMC samples can be found in the appendix.

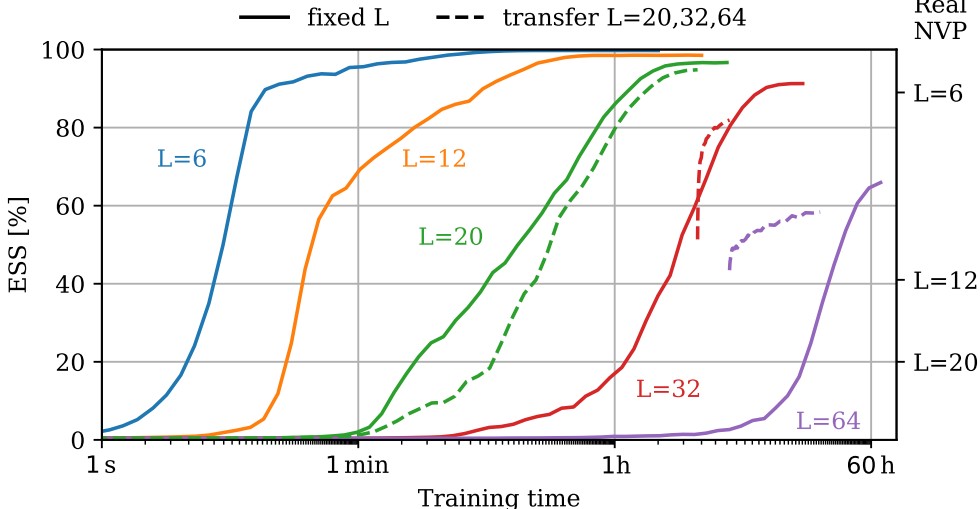

Figure 1: The effective sample sizes for lattice sizes $L \times L$ of our model with coupling constants as in Table 1. Dashed lines show the result of transfer learning from $L = 20$ to $L = 64$ with fixed coupling $\lambda = 4.398$. Ticks on the right indicate the ESS values $89\%, 44\%, 8.3\%, 1.4\%$ achieved with the real NVP architecture for lattice sizes $L = 6, 12, 20, 32$ with coupling constants as in Table 1.

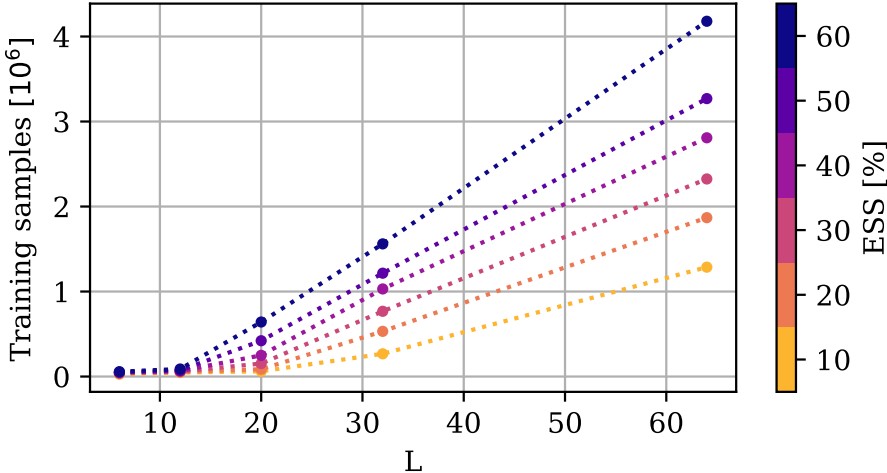

Figure 2: Samples used during training until the values of ESS is attained, for lattice sizes $L = 6, 12, 20, 32, 64$.

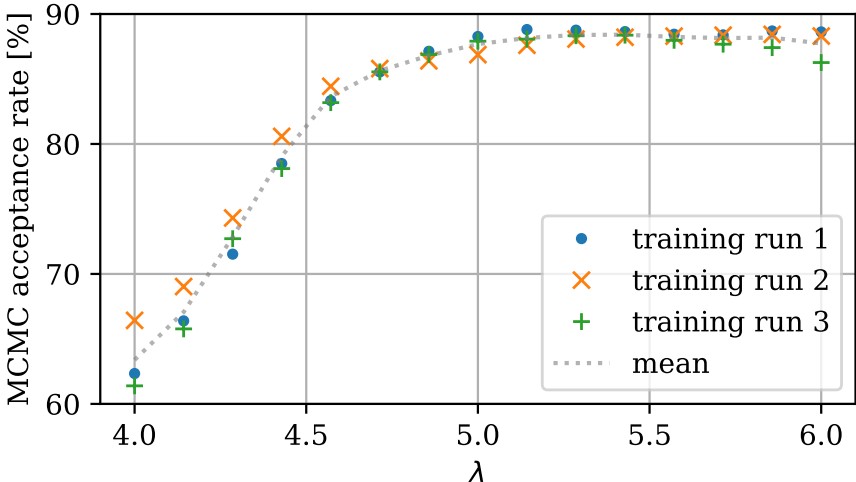

Figure 3: Acceptance rates of MCMC chains of length $10^6$ for a theory-conditional flow trained over a range of $\lambda$ for $L = 32$. The critical coupling is at approximately $\lambda_c = 4.25$.

In addition, we demonstrate that transfer learning as described above can be used to reduce the training time for larger lattices. All three dashed lines in Figure 1 represent a single run of transfer learning between lattice sizes with the target $L = 64$ and the correspondingly fixed value $\lambda = 4.398$. Initially, we train on a theory with $L = 20$ and the $\lambda$ from Table 1 listed for $L = 64$. After the ESS approximately saturates during training, the kernel is scaled to size $32 \times 32$ and training is continued with the same value of the coupling constant $\lambda$ but at $L = 32$. Finally, this is repeated to move to $L = 64$. We observe that, when using transfer learning, it takes a much shorter time to attain a given value of ESS compared to training directly with $L = 64$. To reach an ESS of 50%, for example, it takes about 7.3 hours with transfer learning compared to 36.5 hours without.

## 4.2 Theory-conditional training

In the second experiment, we train the theory-conditional models as summarized in equation (12) over a range of coupling constants from $\lambda = 4$ to $\lambda = 6$. For the model hyperparameters, we now choose $F = 300$, $F' = 20$ for the field basis functions, $D = 21$, $D' = 20$ for the time kernels, and $A = 50$ for the number of $\lambda$ basis expansion functions. The latter are chosen to be of the form

$$J(\lambda)_i = \frac{\exp\left(-w \cdot (\lambda - c_i)^2\right)}{\sum_j \exp\left(-w \cdot (\lambda - c_j)^2\right)}, \tag{16}$$

where $w$ is trainable and the centers $c_i$ are fixed and uniformly spaced in the range of $\lambda$. In order to speed up training, we initially train at $L = 12$ and then transfer to $L = 20$ and finally $L = 32$ as described above. For each size, training was continued until the ESS approximately stabilized. Our model achieves high MCMC acceptance rates between 60% and 90% over the whole range of considered theory space. The robustness of the model was tested by performing the training three times with different random seeds. The achieved performance is similar each run, as can be seen in Figure 3 which shows the MCMC acceptance rates achieved with the setup described above.

To demonstrate the physical usefulness of this approach, Figure 4 shows the susceptibility, computed using the estimator $\chi_1^{(2)}$ via MCMC with the trained model as proposal distribution. With the finite-size scaling taken into account, our numerical results display the expected

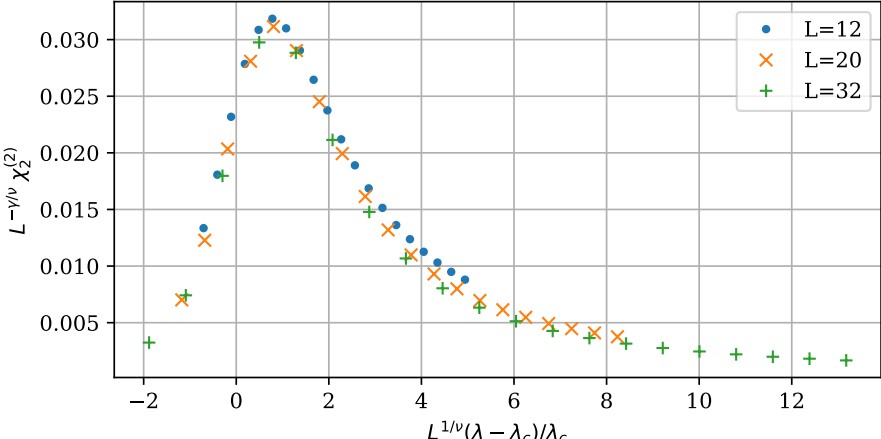

Figure 4: Susceptibility estimated by MCMC chains of length $10^6$, rescaled using the Ising model critical exponents $\nu = 1$, $\gamma = 7/4$ to account for finite-size scaling. The effect of finite lattice sizes can be expressed in terms of a so-called scaling function $g$ as $\chi = L^{\gamma/\nu} g(L^{1/\nu}(\lambda - \lambda_c)/\lambda_c)$ [36]. The used value $\lambda_c = 4.25$ of the critical coupling was chosen based on previous numerical results [34].

collapse of the curves for different lattice sizes. To obtain these results, a theory conditional flow was trained three times for each lattice size with different random seeds over the range of $\lambda$. Using each of the three trained models as proposal, $\chi_1^{(2)}$ was estimated by MCMC. The values shown are the mean while the corresponding standard deviations averaged over $\lambda$ are 0.3%, 0.5%, 1.6% for $L = 12, 20, 32$, respectively, and are thus too small to be visible in the figure.

### 4.3 Symmetry

Figure 5 shows the result of an ablation study we conducted by limiting the symmetries manifestly preserved by the model. Specifically, we relax that the kernels are symmetric with respect to the $D_4$ symmetry. We observe a noticeable reduction of ESS during training. This becomes more significant as the lattice size is increased. We also observed that models without the full symmetries built in are more prone to training instabilities for large lattice sizes.

To investigate the effect of having manifest equivariance compared to approximate equivariance learned during training, we numerically test whether symmetries are broken by the trained real NVP network. The result, shown in Figure 6, shows the quality of the approximation when equivariance of the network is not imposed.

## 5 Conclusion

In this work we propose a fully equivariant continuous flow model as an effective tool for sampling in lattice field theory. Our experiments indicate that such neurally-augmented sampling is a viable method even for large lattices. Moreover, we have shown that the learned parameters can be meaningfully transferred to nearby theories with different correlation lengths, coupling constants, and lattice sizes, even across the critical point as seen in Figure 3. This makes clear that our approach can be employed for tasks where computations need to be done for families of theories, such as phase transition detection and parameter scans. Training can be sped up by transferring between lattice sizes. When ESS per training time is a relevant metric, e.g. in the above cases, the observed trade-off in final ESS achieved after convergence may be acceptable if it cannot be removed by improving the optimization scheme. Practically,

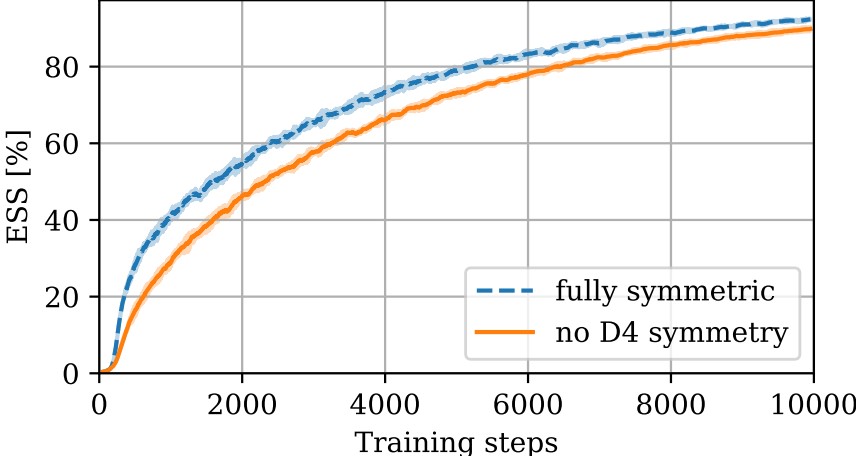

Figure 5: Training performance of the continuous normalizing flow at $L = 20$ with and without enforcing $D4$-symmetry. Shown are the mean and standard deviation (shaded area) over 8 training runs with different random seeds, where the shown value of the ESS is the moving average over 100 training steps.

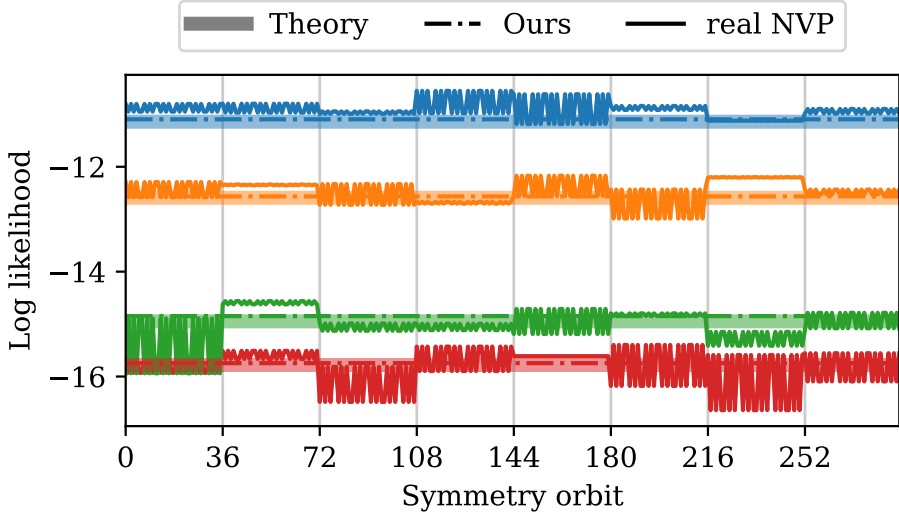

Figure 6: For 4 samples from an MCMC chain with $L = 6$, we show 1) the true log likelihood given by the action, 2) the model log likelihood of our model and 3) real NVP. Log likelihood of 4 samples of an MCMC chain with $L = 6$. The $y$-axis shows the likelihoods when the sample is transformed by all $8 \times 6^2$ symmetries of the lattice.

we note that transferring of parameters can improve training stability for large lattices. Finally, we have demonstrated that our flow model can easily incorporate the geometric and global symmetries of the physical system, resulting in a visible computational advantage which increases with the lattice size. These developments make our flow-model approach a practical tool ready to be employed in lattice field theory computations.

In future work, it will be interesting to study how the methods explored in this paper can be applied to other theories beyond single scalar fields, and in particular gauge theories. While methods such as parameter transfer and conditioned networks may immediately be employed, more work is needed to define appropriately equivariant models. We note that continuous normalizing flows have now been applied to gauge theories [37], as a machine-learning generalization of Lüscher's original trivializing maps [24].

We will end with a comment on a potential interpretation of the learned ODE flow. Note that the flow of the density described by the ODE (8) can readily be interpreted as a flow in the space $\mathcal{M}$ of coupling constants by identifying the probability distribution $p(t)$ as the Boltzmann distribution of a Hamiltonian $H_t$. As such, it is tantalizing to try to understand the flow in terms of the physical renormalization group of the theory, which also renders a flow in $\mathcal{M}$. We will report on this in the future.

## Acknowledgments

We wish to thank Max Welling for inspiring discussions, Vassilis Anagiannis for his contribution in the early stages of this work, and Jonas Köhler for his helpful suggestions. We also thank Joe Marsh Rossney, Luigi Del Debbio, Kim Nicoli, and Uroš Seljak for helpful correspondence.

## A  Observables

In the MCMC setting, observables are estimated using a set of $N$ samples $\left\{\phi^{(i)}\right\}_{i=1}^{N}$ from the target distribution. These are generated here by applying the Metropolis-Hasting step on samples proposed by the trained continuous normalizing flows. Here we consider scalar fields on a two-dimensional periodic square lattice of length $L$, $V_L \cong (\mathbb{Z}/L\mathbb{Z})^2$.

To compute the correlation length $\xi$ we first introduce the estimator for the connected two-point Green's function

$$G(x) = \frac{1}{L^2} \sum_{y \in V_L} \frac{1}{N} \sum_{i=1}^{N} \left[ \phi_y^{(i)} \phi_{x+y}^{(i)} - \phi_y^{(i)} \frac{1}{N} \sum_{j=1}^{N} \phi_{x+y}^{(j)} \right]. \tag{A.1}$$

Summing over one of the lattice dimensions, we have

$$G_s(x_2) = \sum_{x_1=1}^{L} G(x = (x_1, x_2)). \tag{A.2}$$

The correlation length can then be estimated via

$$\frac{1}{\xi} = \frac{1}{L-1} \sum_{x_2=1}^{L-1} \operatorname{arcosh}\left( \frac{G_s(x_2+1) + G_s(x_2-1)}{2G_s(x_2)} \right). \tag{A.3}$$

The two-point susceptibility is estimated as

$$\chi_2^{(1)} = \sum_{x \in V_L} G(x) = L^2 \left( \frac{1}{N} \sum_{i=1}^{N} \overline{\phi^{(i)}}^2 - \left( \frac{1}{N} \sum_{i=1}^{N} \overline{\phi^{(i)}} \right)^2 \right),$$

$$\text{where} \quad \overline{\phi^{(i)}} = \frac{1}{L^2} \sum_{x \in V_L} \phi_x^{(i)}. \tag{A.4}$$

In the broken phase, the distribution of $\overline{\phi}$ becomes bimodal. To address this, we introduce an alternative estimator for the two-point susceptibility [38]

$$\chi_2^{(2)} = L^2 \left( \frac{1}{N} \sum_{i=1}^{N} \overline{\phi^{(i)}}^2 - \left( \frac{1}{N} \sum_{i=1}^{N} \left| \overline{\phi^{(i)}} \right| \right)^2 \right). \tag{A.5}$$

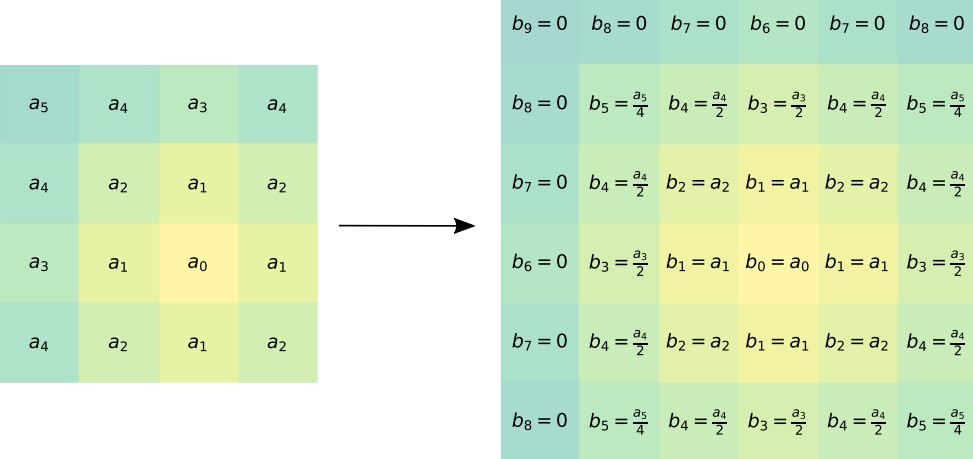

Figure 7: Illustration of transferring weights $W$ between lattices of size $L = 4$ and $L' = 6$.

In the table of estimated observables at different lattice sizes, the ESS was computed 100 times using different random seeds, each time using 500 samples. MCMC of length $10^6$ was run 10 times with different random seeds. In both cases, the average and its estimated error are shown. The observables were computed with a single MCMC run of length $10^6$ and errors estimated using bootstrap drawing 100 times and with bin size 4. A thermalization transient of length $10^5$ was discarded.

## B   Experimental details

### Network architecture

The width factor in the chosen form of the $\lambda$ basis expansion functions $J(\lambda)_i$ should not become negative during training. It is thus parametrized by the trainable parameter $\tilde{w}$ as

$$w = (A-1)\log(1 + \exp(\tilde{w})), \tag{B.1}$$

which is initialized to $\tilde{w} = \log(e-1)$. $A$ denotes the number of $\lambda$ basis expansion function The frequencies $\omega_f$ are initialised by random uniform values between 0 and 5. Orthogonal initialization was used for the matrices $W^K$ and $W^H$, while $\tilde{W}$ is initially zero.

An example illustrating how weights are upscaled to a larger lattice size can be found in Figure 7.

### Loss computation

The KL-divergence between the network's output distribution and the theory distribution is computed by a Monte Carlo approximation. For the initial flow model, 256 samples are used in each training step to approximate the integral. To train the theory-conditional flows, a sampling scheme for the considered theory parameters must be chosen. Here, in each training step 8 different values of $\lambda$ are sampled uniformly in the chosen range. For each, the KL-divergence is computed by Monte Carlo approximation with 128 samples. The loss is then the average of the 8 different KL-divergences. The training performance was not observed to strongly depend on the chosen sample sizes. The above values are chosen as a compromise based on available computational resources and are not the result of a comprehensive hyperparameter search.

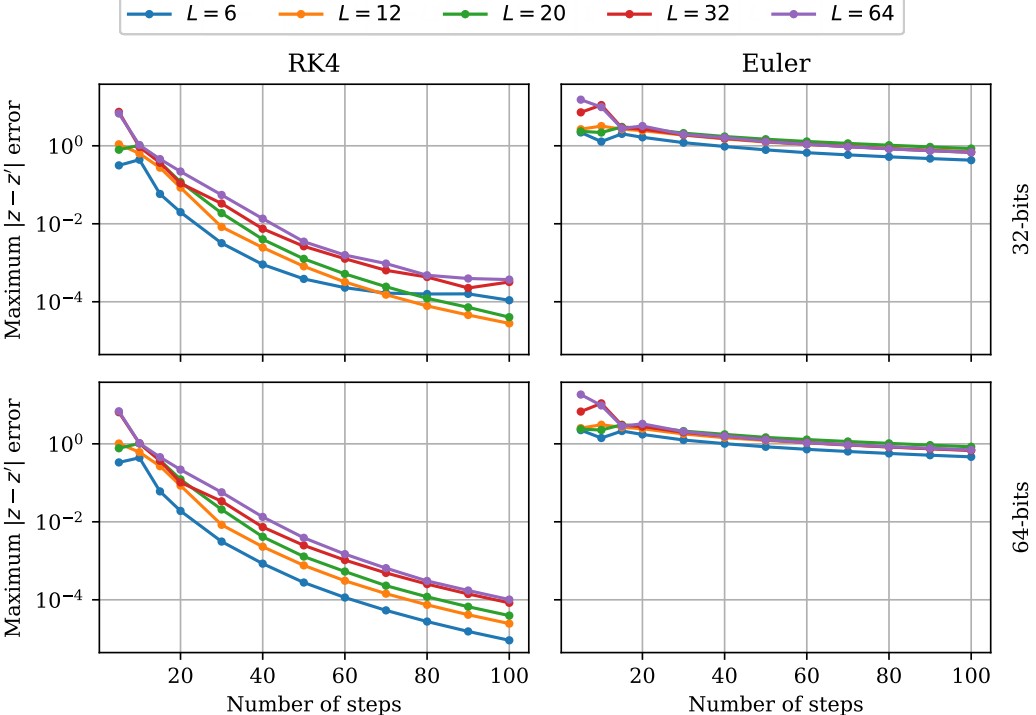

Figure 8: Maximum discretization error as measured by the deviation of a chained forward and reverse flow of the single-theory model from the identity. Parameters of the network for each lattice size are those obtained in the numerical experiments of section 4.1. For each number of steps, the reported maximum error was computed using 4096 samples.

**Optimizer**

The flow networks are optimized by minimizing the KL-divergence using stochastic gradient descent. For this, the Adam optimizer is used with a learning rate of 0.005, decaying exponentially by a factor of 0.01 every 8000 steps. After transferring to a larger lattice, the initial learning rate was reduced to $10^{-5}$. The decay parameters $\beta_1 = 0.8$, and $\beta_2 = 0.9$ of the Adam optimizer were found to improve training speed. The same values were used for all lattice sizes. Training efficiency can likely be improved by tuning these hyperparameters.

## C Discretization error

An important consideration when employing continuous normalizing flows as a sampling method is the impact of the discretization error introduced by solving the defining ODE numerically. Generally, this becomes significant if the discretization error is more significant, in magnitude, than the statistical error inherent to the Monte Carlo approximation. Fortunately, once training is done, the proposal generation itself can be done in parallel, which means that it is not the primary bottleneck. This increases the freedom in choosing the (possibly black-box) integrator. If the integration accuracy is insufficient, one may:

- Increase the number of steps in the discretization.

- Choose another (higher order) integration method.

- Increase the floating point accuracy from 32 to 64-bits.

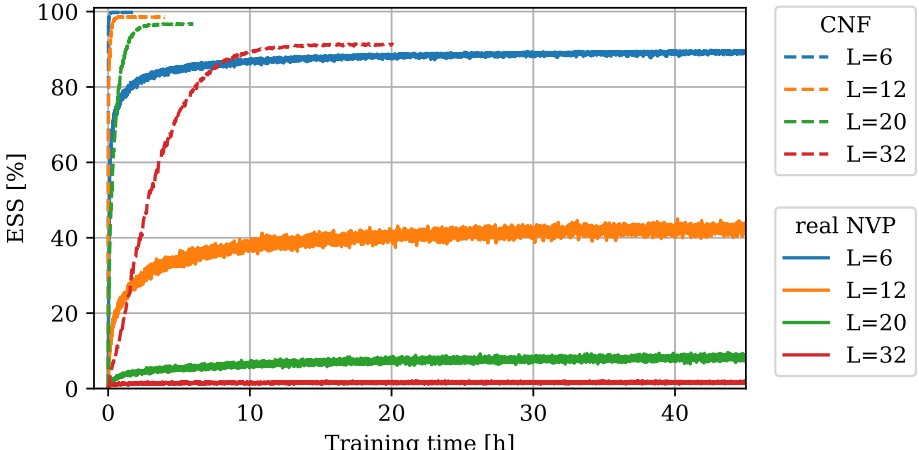

Figure 9: ESS achieved by our continuous normalizing flow model compared to the real NVP model as described in [14] for different lattice sizes in relation to training time on the same machine.

In general, these choices do not have to be the same as those used during training. It may also be advantageous to increase the integration accuracy after some amount of training.

One measure to study the discretization error numerically is the difference between a composition of forward and reverse flow and the identity. Specifically, if we sample $z \sim \mathcal{N}(0, \mathbf{1})$ then $z' = (f^{-1} \circ f)(z)$ should equal $z$ for any learned normalizing flow $f$. Crucially, this does not depend on the flow having learned the theory correctly, and we need not worry about relative magnitudes as, by definition, $z \sim O(1)$. Any deviation of $z - z'$ from 0 measures the error of the numerical integrator.

Figure 8 shows the maximum of $|z - z'|$, both over sites and samples, for the RK4 and the much simpler Euler integrator over a range of integration steps. Note that the integration time is fixed to $0 \le t \le 1$, so setting the number of steps is equivalent to choosing the step sizes. As expected, the integration error reduces if the number of steps is increased. The difference between the two integration methods is notable, with the higher order RK4 showing a significantly faster reduction of error.

The figure shows results both for 32-bits (as used for training and in the main part of the paper) and 64-bits floating point precision. Due to floating point errors, the integration error saturates after some number of steps, as expected. This appears to occur around 60 steps in the case of RK4 with 32-bits. By increasing precision to 64-bits, the number of steps can be increased to further reduce the error below $10^{-4}$. In other experiments, we have found that this error can be further reduced using higher order methods. In particular, using the Tsit5 integration method and 64-bits precision we found the error at 100 steps to be around $10^{-6}$.

# D Comparison to real NVP

As a benchmark, we compare the ESS achieved by our model with that achieved by the real NVP architecture [14]. The results for the real NVP model were initially obtained with an implementation in PyTorch, with training until the ESS approximately stabilizes. To have a more direct comparison than just running the code on the same machine, we also implemented the code in JAX, the same framework used for our continuous flow model, which led to a faster run-time per training step. Figure 9 shows the ESS achieved by each model in relation to training time, using the faster JAX run time obtained on a single NVIDIA Titan RTX GPU.

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
