# Peer review of "Learning Lattice Quantum Field Theories with Equivariant Continuous Flows"

_SciPost Physics, doi:SciPost Phys. 15, 238 (2023)_

## Round 1 · Referee Report · Anonymous (Referee 1) · 2023-5-29

Strengths

1) Well-motivated 2) Of interest 3) Relatively original 4) Robust

Weaknesses

1) Minor corrections are necessary

Report

The submitted article presents a machine learning model to sample lattice field theories. The main goal of the work is to try and tackle the scalability challenges that still affect the sampling of lattice field theory configuration space using machine learning methods. The idea proposed herein is based on a continuous version of trivializing flow models (relying on a neural ordinary differential equation) and is tested in scalar field theory in two dimensions.
As compared to similar previous studies, the model presented here offers a better treatment of the symmetries of the theory (following ideas that have been previously applied for molecular modelling) and leads to an improved effective sample size even for relatively large lattices.
The work is well motivated and the topic is of interest. The idea combines existing methods in an original way and the numerical work appears to be robust. The manuscript is well written and the bibliography is quite complete.
The manuscript meets this journal´s acceptance criteria.
I request some corrections, which are included in the ´´Requested changes´´ list.

Requested changes

Requested corrections include the following: 1) On page 3, write that the explicit definition of the correlation length is given in the appendix. 2) On page 3, just before the beginning of Subsection 2.1, add some details about the CFTs that are expected to describe the critical point of the model, in particular depending on the degree of the potential. 3) Page 5: have different function bases H (among those that respect the symmetries of the problem and are sufficiently analytically tractable; for example, some families of polynomials or special functions?) been tested? If so, does the Fourier basis they use perform better? 4) Page 6: a couple of lines above Eq.(12): how is A defined? 5) Page 7: I would recommend replacing the word ´´experiment´´ with ´´numerical test´´ in the title of Section 4 and throughout the rest of the paper. 6) Page 9: second line: perhaps a word is missing in ´´by performing the training three runs´´? (Maybe the sentence was meant to be ´´by performing the training in three runs´´?) 7) Page 9: figure 4: the data appear to collapse on a common curve: have the authors tried to fit it to some known function? 8) Page 10, fifth line in Section 5: ´´even across the critical point´´: is it really so? Can the authors add more results or at least more comments on this? 9) Page 10, in Section 5, it would be interesting to know if the approach used in this work can be generalised to local symmetries, that is to gauge theories. It would be helpful if a discussion on this issue could be included in the Conclusions. 10) Page 11: last sentence of appendix A: replace ´´burn-in phase´´ with ´´thermalisation transient´´. 11) Page 14: add journal publication details for Reference 13: Phys.Rev.D 107 (2023) 5, 054501. 12) Page 14: in Reference 15 correct ´´Racaniere´´ with ´´Racanière´´. 13) Page 14: add journal publication details for Reference 20: PoS LATTICE2022 (2023) 036. 14) Page 15: add hyperlink for Reference 24. 15) Page 15: in Reference 31 correct ´´Noe´´ with ´´Noé´´. 16) Page 15: add hyperlink for Reference 32.

---

## Round 1 · Referee Report · Anonymous (Referee 2) · 2023-6-29

Strengths

  • The use of continuous flows to generate the field transformations, preserving spatial symmetries
  • The possibility of learning a family of models that parametrize different values of the parameters present in the action.
  • Better performance in the trainning.

Weaknesses

1) Claims about better scalability of the trainning are not motivated (and probably not true). 2) Invertibility of their transformations at finite precision arithmetic has to be shown.

Report

This paper present an important contribution to a very active área of research in Lattice QCD: the possibility of using Machine Learning (ML) techniques to samples of the Lattice Field Theories. The paper presents some novel ingredients compared with previously published works:

  • The use of continuous flows to generate the field transformations, preserving spatial symmetries
  • The possibility of learning a family of models that parametrize different values of the parameters present in the action.

I find these ingredients very interesting. The paper is well written and in my opinion meets the criteria for publishing. Nevertheless there are two points that I think that have to be either explained better or corrected.

First, the paper insists that with their network architecture and strategy (using as change of variables the solution of the ODE), the scaling with the lattice size, measured using the ESS, improves dramatically. But this point is very difficult to understand. Given that the network architecture fully exploits invariance under translations, and that the ESS is the exponential of an extensive quantity (the difference of actions), it seems unavoidable that for asymptotically large volumes, the ESS will deteriorate exponentially fast at fixed training. I think that the authors have nicely shown that their training is much more efficient than what has been found in previous works. Still the numerical evidence (L/a=32 in two dimensions) is really far from the interesting cases of the Lattice QCD regime.

Second, there is a delicate numerical accuracy issue in their construction. The type of flow equation that the authors use are most probably parabolic in nature. It is well known that backwards parabolic equations are not well-behaved. Numerical solutions may fail to converge and/or the convergence might strongly depend on the integrator used (it is also understood that high order Runge-Kutta schemes are particularly bad). Note that this numerical instabilities might arise for some values of the parameters that define the neural network. For these reasons I think that it is important to include the results of an experiment, where the integration forward in time is performed, and then the integration backwards in time, and a measure of the difference between the initial value of the parameters is presented.

In summary I think that the statements about a better scaling might need to be softened a bit (or more evidence needs to be provided). Moreover some tests that invertibility of their transformation is preserved at finite precision arithmetic has to be presented.

On more general grounds, their approach share some ideas with the work of M. Luscher "Trivializing maps, the Wilson flow and the HMC algorithm". In the mentioned work the transformation, also based on a very similar ODE, was constructed analytically instead of by numerical methods, but still I think that a reference to this work should be added to the text.

Once these two issues are addressed in the manuscript I will be very happy to recommend the work for publication.

Requested changes

1) statements about a better scaling might need to be softened a bit (or more evidence needs to be provided) 2) tests that invertibility of their transformation is preserved at finite precision arithmetic has to be presented. 3) Reference seminal work by M. Luscher "Trivializing maps, the Wilson flow and the HMC algorithm".

---

## Round 2 · Referee Report · Anonymous (Referee 1) · 2023-10-8

Report

The submitted article was revised as requested. The new version is accepted for publication.

---

## Round 2 · Author Response

We want to thank the referees for carefully reviewing our manuscript and providing helpful suggestions. We hope to have incorporated the feedback in our modifications, as summarized in the list of changes.

---

## Round 2 · List of Changes

• Mention M. Lüscher's "Trivializing maps, the Wilson flow and the HMC algorithm" in the introduction.
  • Removed claim about exponential scaling of training cost in lattice size.
  • Slight rewording for clarity and added reference to the appendix in the introduction of section 2.
  • In section 3.1, added a remark on possible choices for basis functions besides the trigonometric ones.
  • Change the section 4 title from "Experiments" to "Numerical Tests", as suggested by the referee.
  • At the beginning of section 4, added a remark about the integration method and reference to the appendix for discussion of the discretization error.
  • Mention the location of the critical point in the caption of Figure 3.
  • Some changes in word choice, following referee suggestions ("thermalization transient" instead of "burn-in phase", "performing the training three times" instead of "runs").
  • In the conclusion, added a reference to Figure 3 to substantiate the claim that training can be performed over coupling values crossing the critical point. Also added a comment on how this work may be extended to other theories and a reference to related work.
  • Added a section to the appendix that evaluates and discusses the discretization error for our trained models. We include different choices of step sizes, integration methods (Euler and RK4), and 32/64-bit precisions.
  • Publication updates and typographical corrections in the list of references.
  • Correction of a typographical mistake in the list of institutional affiliations (one footnote was erroneously listed as an association).

---

## Editorial Decision

published